# Differentially Private Empirical Risk Minimization Revisited: Faster and More General*

**Di Wang**
Dept. of Computer Science and Engineering
State University of New York at Buffalo
Buffalo, NY 14260
dwang45@buffalo.edu

**Minwei Ye**
Dept. of Computer Science and Engineering
State University of New York at Buffalo
Buffalo, NY 14260
minweiye@buffalo.edu

**Jinhui Xu**
Dept. of Computer Science and Engineering
State University of New York at Buffalo
Buffalo, NY 14260
jinhui@buffalo.edu

## Abstract

In this paper we study the differentially private Empirical Risk Minimization (ERM) problem in different settings. For smooth (strongly) convex loss function with or without (non)-smooth regularization, we give algorithms that achieve either optimal or near optimal utility bounds with less gradient complexity compared with previous work. For ERM with smooth convex loss function in high-dimensional ($p \gg n$) setting, we give an algorithm which achieves the upper bound with less gradient complexity than previous ones. At last, we generalize the expected excess empirical risk from convex loss functions to non-convex ones satisfying the Polyak-Lojasiewicz condition and give a tighter upper bound on the utility than the one in [34].

## 1 Introduction

Privacy preserving is an important issue in learning. Nowadays, learning algorithms are often required to deal with sensitive data. This means that the algorithm needs to not only learn effectively from the data but also provide a certain level of guarantee on privacy preserving. Differential privacy is a rigorous notion for statistical data privacy and has received a great deal of attentions in recent years [11, 10]. As a commonly used supervised learning method, Empirical Risk Minimization (ERM) also faces the challenge of achieving simultaneously privacy preserving and learning. Differentially Private (DP) ERM with convex loss function has been extensively studied in the last decade, starting from [7]. In this paper, we revisit this problem and present several improved results.

**Problem Setting** Given a dataset $D = \{z_1, z_2 \cdots, z_n\}$ from a data universe $\mathcal{X}$, and a closed convex set $\mathcal{C} \subseteq \mathbb{R}^p$, DP-ERM is to find

$$x_* \in \arg\min_{x \in \mathcal{C}} F^r(x, D) = F(x, D) + r(x) = \frac{1}{n} \sum_{i=1}^{n} f(x, z_i) + r(x)$$

with the guarantee of being differentially private. We refer to $f$ as loss function. $r(\cdot)$ is some simple (non)-smooth convex function called regularizer. If the loss function is convex, the utility of the

| | Method | Utility Upper Bd | Gradient Complexity | Non smooth Regularizer? |
|---|---|---|---|---|
| [8][7] | Objective Perturbation | $O(\frac{p}{n^2\epsilon^2})$ | N/A | No |
| [21] | Objective Perturbation | $O(\frac{p}{n^2\epsilon^2} + \frac{\lambda\|x_*\|^2}{n\epsilon})$ | N/A | Yes |
| [6] | Gradient Perturbation | $O(\frac{p\log^2(n)}{n^2\epsilon^2})$ | $O(n^2)$ | Yes |
| [34] | Output Perturbation | $O(\frac{p}{n^2\epsilon^2})$ | $O(n\kappa\log(\frac{n\epsilon}{\kappa}))$ | No |
| **This Paper** | Gradient Perturbation | $O(\frac{p\log(n)}{n^2\epsilon^2})$ | $O((n+\kappa)\log(\frac{n\epsilon\mu}{p}))$ | Yes |

Table 1: Comparison with previous $(\epsilon, \delta)$-DP algorithms. We assume that the loss function $f$ is convex, 1-smooth, differentiable (twice differentiable for objective perturbation), and 1-Lipschitz. $F^r$ is $\mu$-strongly convex. Bound and complexity ignore multiplicative dependence on $\log(1/\delta)$. $\kappa = \frac{L}{\mu}$ is the condition number. The lower bound is $\Omega(\min\{1, \frac{p}{n^2\epsilon^2}\})$[6].

algorithm is measured by the expected excess empirical risk, *i.e.* $\mathbb{E}[F^r(x^{\text{private}}, D)] - F^r(x_*, D)$. The expectation is over the coins of the algorithm.

A number of approaches exist for this problem with convex loss function, which can be roughly classified into three categories. The first type of approaches is to perturb the output of a non-DP algorithm. [7] first proposed output perturbation approach which is extended by [34]. The second type of approaches is to perturb the objective function [7]. We referred to it as objective perturbation approach. The third type of approaches is to perturb gradients in first order optimization algorithms. [6] proposed gradient perturbation approach and gave the lower bound of the utility for both general convex and strongly convex loss functions. Later, [28] showed that this bound can actually be broken by adding more restrictions on the convex domain $\mathcal{C}$ of the problem.

As shown in the following tables[2] , the output perturbation approach can achieve the optimal bound of utility for strongly convex case. But it cannot be generalized to the case with non-smooth regularizer. The objective perturbation approach needs to obtain the optimal solution to ensure both differential privacy and utility, which is often intractable in practice, and cannot achieve the optimal bound. The gradient perturbation approach can overcome all the issues and thus is preferred in practice. However, its existing results are all based on Gradient Descent (GD) or Stochastic Gradient Descent (SGD). For large datasets, they are slow in general. In the first part of this paper, we present algorithms with tighter utility upper bound and less running time. Almost all the aforementioned results did not consider the case where the loss function is non-convex. Recently, [34] studied this case and measured the utility by gradient norm. In the second part of this paper, we generalize the expected excess empirical risk from convex to Polyak-Lojasiewicz condition, and give a tighter upper bound of the utility given in [34]. Due to space limit, we leave many details, proofs, and experimental studies in the supplement.

## 2 Related Work

There is a long list of works on differentially private ERM in the last decade which attack the problem from different perspectives. [17][30] and [2] investigated regret bound in online settings. [20] studied regression in incremental settings. [32] and [31] explored the problem from the perspective of learnability and stability. We will compare to the works that are most related to ours from the utility and gradient complexity (*i.e.,* the number (complexity) of first order oracle $(f(x, z_i), \nabla f(x, z_i))$ being called) points of view. **Table 1** is the comparison for the case that loss function is strongly convex and 1-smooth. Our algorithm achieves near optimal bound with less gradient complexity compared with previous ones. It is also robust to non-smooth regularizers.

**Tables 2 and 3** show that for non-strongly convex and high-dimension cases, our algorithms outperform other peer methods. Particularly, we improve the gradient complexity from $O(n^2)$ to $O(n\log n)$ while preserving the optimal bound for non-strongly convex case. For high-dimension case, gradient complexity is reduced from $O(n^3)$ to $O(n^{1.5})$. Note that [19] also considered high-dimension case

| | Method | Utility Upper Bd | Gradient Complexity | Non smooth Regularizer? |
|---|---|---|---|---|
| [21] | Objective Perturbation | $O(\frac{\sqrt{p}}{n\epsilon})$ | N/A | Yes |
| [6] | Gradient Perturbation | $O(\frac{\sqrt{p}\log^{3/2}(n)}{n\epsilon})$ | $O(n^2)$ | Yes |
| [34] | Output Perturbation | $O([\frac{\sqrt{p}}{n\epsilon}]^{\frac{2}{3}})$ | $O(n[\frac{n\epsilon}{d}]^{\frac{2}{3}})$ | No |
| **This paper** | Gradient Perturbation | $O(\frac{\sqrt{p}}{n\epsilon})$ | $O(\frac{n\epsilon}{\sqrt{p}} + n\log(\frac{n\epsilon}{p}))$ | Yes |

Table 2: Comparison with previous $(\epsilon, \delta)$-DP algorithms, where $F^r$ is not necessarily strongly convex. We assume that the loss function $f$ is convex, 1-smooth, differentiable( twice differentiable for objective perturbation), and 1-Lipschitz. Bound and complexity ignore multiplicative dependence on $\log(1/\delta)$. The lower bound in this case is $\Omega(\min\{1, \frac{\sqrt{p}}{n\epsilon}\})$[6].

via dimension reduction. But their method requires the optimal value in the dimension-reduced space, in addition they considered loss functions under the condition rather than $\ell_2$- norm Lipschitz.

For non-convex problem under differential privacy, [15][9][13] studied private SVD. [14] investigated k-median clustering. [34] studied ERM with non-convex smooth loss functions. In [34], the authors defined the utility using gradient norm as $\mathbb{E}[||\nabla F(x^{\text{private}})||^2]$. They achieved a qualified utility in $O(n^2)$ gradient complexity via DP-SGD. In this paper, we use DP-GD and show that it has a tighter utility upper bound.

| | Method | Utility Upper Bd | Gradient Complexity | Non smooth Regularizer? |
|---|---|---|---|---|
| [28] | Gradient Perturbation | $O(\frac{\sqrt{G_{\mathcal{C}}^2+||\mathcal{C}||^2}\log(n)}{n\epsilon})$ | $O(\frac{n^3\epsilon^2}{(G_{\mathcal{C}}^2+||\mathcal{C}||^2)\log^2(n)})$ | Yes |
| [28] | Objective Perturbation | $O(\frac{G_{\mathcal{C}}+\lambda||\mathcal{C}||^2}{n\epsilon})$ | N/A | No |
| [29] | Gradient Perturbation | $O(\frac{(G_{\mathcal{C}}^{\frac{2}{3}}\log^2(n))}{(n\epsilon)^{\frac{2}{3}}})$ | $O(\frac{(n\epsilon)^{\frac{2}{3}}}{G_{\mathcal{C}}^{\frac{2}{3}}})$ | Yes |
| **This paper** | Gradient Perturbation | $O(\frac{\sqrt{G_{\mathcal{C}}^2+||\mathcal{C}||^2}}{n\epsilon})$ | $O\left(\frac{n^{1.5}\sqrt{\epsilon}}{(G_{\mathcal{C}}^2+||\mathcal{C}||^2)^{\frac{1}{4}}}\right)$ | No |

Table 3: Comparison with previous $(\epsilon, \delta)$-DP algorithms. We assume that the loss function $f$ is convex, 1-smooth, differentiable( twice differentiable for objective perturbation), and 1-Lipschitz. The utility bound depends on $G_{\mathcal{C}}$, which is the Gaussian width of $\mathcal{C}$. Bound and complexity ignore multiplicative dependence on $\log(1/\delta)$.

# 3 Preliminaries

**Notations:** We let $[n]$ denote $\{1, 2, \ldots, n\}$. Vectors are in column form. For a vector $v$, we use $||v||_2$ to denote its $\ell_2$-norm. For the gradient complexity notation, $G, \delta, \epsilon$ are omitted unless specified. $D = \{z_1, \cdots, z_n\}$ is a dataset of n individuals.

**Definition 3.1** (Lipschitz Function over $\theta$)**.** A loss function $f : \mathcal{C} \times \mathcal{X} \to \mathbb{R}$ is G-Lipschitz (under $\ell_2$-norm) over $\theta$, if for any $z \in \mathcal{X}$ and $\theta_1, \theta_2 \in \mathcal{C}$, we have $|f(\theta_1, z) - f(\theta_2, z)| \leq G||\theta_1 - \theta_2||_2$.

**Definition 3.2** (L-smooth Function over $\theta$)**.** A loss function $f : \mathcal{C} \times \mathcal{X} \to \mathbb{R}$ is L-smooth over $\theta$ with respect to the norm $|| \cdot ||$ if for any $z \in \mathcal{X}$ and $\theta_1, \theta_2 \in \mathcal{C}$, we have

$$||\nabla f(\theta_1, z) - \nabla f(\theta_2, z)||_* \leq L||\theta_1 - \theta_2||,$$

where $|| \cdot ||_*$ is the dual norm of $|| \cdot ||$. If $f$ is differentiable, this yields

$$f(\theta_1, z) \leq f(\theta_2, z) + \langle \nabla f(\theta_2, z), \theta_1 - \theta_2 \rangle + \frac{L}{2}||\theta_1 - \theta_2||^2.$$

We say that two datasets $D, D'$ are neighbors if they differ by only one entry, denoted as $D \sim D'$.

**Definition 3.3** (Differentially Private[11])**.** A randomized algorithm $\mathcal{A}$ is $(\epsilon, \delta)$-differentially private if for all neighboring datasets $D, D'$ and for all events $S$ in the output space of $\mathcal{A}$, we have

$$Pr(\mathcal{A}(D) \in S) \leq e^{\epsilon} Pr(\mathcal{A}(D') \in S) + \delta,$$

when $\delta = 0$ and $\mathcal{A}$ is $\epsilon$-differentially private.

We will use Gaussian Mechanism [11] and moments accountant [1] to guarantee $(\epsilon, \delta)$-DP.

**Definition 3.4** (Gaussian Mechanism). Given any function $q : \mathcal{X}^n \to \mathbb{R}^p$, the Gaussian Mechanism is defined as:

$$\mathcal{M}_G(D, q, \epsilon) = q(D) + Y,$$

where Y is drawn from Gaussian Distribution $\mathcal{N}(0, \sigma^2 I_p)$ with $\sigma \geq \frac{\sqrt{2\ln(1.25/\delta)}\Delta_2(q)}{\epsilon}$. Here $\Delta_2(q)$ is the $\ell_2$-sensitivity of the function $q$, i.e. $\Delta_2(q) = \sup_{D \sim D'} ||q(D) - q(D')||_2$. Gaussian Mechanism preservers $(\epsilon, \delta)$-differentially private.

The moments accountant proposed in [1] is a method to accumulate the privacy cost which has tighter bound for $\epsilon$ and $\delta$. Roughly speaking, when we use the Gaussian Mechanism on the (stochastic) gradient descent, we can save a factor of $\sqrt{\ln(T/\delta)}$ in the asymptotic bound of standard deviation of noise compared with the advanced composition theorem in [12].

**Theorem 3.1** ([1]). For $G$-Lipschitz loss function, there exist constants $c_1$ and $c_2$ so that given the sampling probability $q = l/n$ and the number of steps T, for any $\epsilon < c_1 q^2 T$, a DP stochastic gradient algorithm with batch size $l$ that injects Gaussian Noise with standard deviation $\frac{G}{n}\sigma$ to the gradients (Algorithm 1 in [1]), is $(\epsilon, \delta)$-differentially private for any $\delta > 0$ if

$$\sigma \geq c_2 \frac{q\sqrt{T\ln(1/\delta)}}{\epsilon}.$$

# 4 Differentially Private ERM with Convex Loss Function

In this section we will consider ERM with (non)-smooth regularizer[3], i.e.

$$\min_{x \in \mathbb{R}^p} F^r(x, D) = F(x, D) + r(x) = \frac{1}{n}\sum_{i=1}^{n} f(x, z_i) + r(x). \tag{1}$$

The loss function $f$ is convex for every $z$. We define the proximal operator as

$$\text{prox}_r(y) = \arg\min_{x \in \mathbb{R}^p}\{\frac{1}{2}||x - y||_2^2 + r(x)\},$$

and denote $x_* = \arg\min_{x \in \mathbb{R}^p} F^r(x, D)$.

---

**Algorithm 1** DP-SVRG($F^r, \tilde{x}_0, T, m, \eta, \sigma$)

**Input**: $f(x, z)$ is G-Lipschitz and L-smooth. $F^r(x, D)$ is $\mu$-strongly convex w.r.t $\ell_2$-norm. $\tilde{x}_0$ is the initial point, $\eta$ is the step size, $T, m$ are the iteration numbers.

1: **for** $s = 1, 2, \cdots, T$ **do**
2:      $\tilde{x} = \tilde{x}_{s-1}$
3:      $\tilde{v} = \nabla F(\tilde{x})$
4:      $x_0^s = \tilde{x}$
5:      **for** $t = 1, 2, \cdots, m$ **do**
6:          Pick $i_t^s \in [n]$
7:          $v_t^s = \nabla f(x_{t-1}^s, z_{i_t^s}) - \nabla f(\tilde{x}, z_{i_t^s}) + \tilde{v} + u_t^s$, where $u_t^s \sim \mathcal{N}(0, \sigma^2 I_p)$
8:          $x_t^s = \text{prox}_{\eta r}(x_{t-1}^s - \eta v_t^s)$
9:      **end for**
10:     $\tilde{x}_s = \frac{1}{m}\sum_{k=1}^{m} x_k^s$
11: **end for**
12: **return** $\tilde{x}_T$

## 4.1 Strongly convex case

We first consider the case that $F^r(x, D)$ is $\mu$-strongly convex, **Algorithm 1** is based on the Prox-SVRG [33], which is much faster than SGD or GD. We will show that DP-SVRG is also faster than DP-SGD or DP-GD in terms of the time needed to achieve the near optimal excess empirical risk bound.

**Definition 4.1** (Strongly Convex). The function $f(x)$ is $\mu$-strongly convex with respect to norm $||\cdot||$ if for any $x, y \in \text{dom}(f)$, there exist $\mu > 0$ such that

$$f(y) \geq f(x) + \langle \partial f, y - x \rangle + \frac{\mu}{2}||y - x||^2, \tag{2}$$

where $\partial f$ is any subgradient on $x$ of $f$.

**Theorem 4.1.** In **DP-SVRG**(Algorithm 1), for $\epsilon \leq c_1 \frac{Tm}{n^2}$ with some constant $c_1$ and $\delta > 0$, it is $(\epsilon, \delta)$-differentially private if

$$\sigma^2 = c\frac{G^2 Tm \ln(\frac{1}{\delta})}{n^2 \epsilon^2} \tag{3}$$

for some constant $c$.

**Remark 4.1.** The constraint on $\epsilon$ in Theorems 4.1 and 4.3 comes from Theorem 3.1. This constraint can be removed if the noise $\sigma$ is amplified by a factor of $O(\ln(T/\delta))$ in (3) and (6). But accordingly there will be a factor of $\tilde{O}(\log(Tm/\delta))$ in the utility bound in (5) and (7). In this case the guarantee of differential privacy is by advanced composition theorem and privacy amplification via sampling[6].

**Theorem 4.2** (Utility guarantee). Suppose that the loss function $f(x, z)$ is convex, G-Lipschitz and L-smooth over $x$. $F^r(x, D)$ is $\mu$-strongly convex w.r.t $\ell_2$-norm. In **DP-SVRG**(Algorithm 1), let $\sigma$ be as in (3). If one chooses $\eta = \Theta(\frac{1}{L}) \leq \frac{1}{12L}$ and sufficiently large $m = \Theta(\frac{L}{\mu})$ so that they satisfy inequality

$$\frac{1}{\eta(1 - 8\eta L)\mu m} + \frac{8L\eta(m+1)}{m(1 - 8L\eta)} < \frac{1}{2}, \tag{4}$$

then the following holds for $T = O\left(\log(\frac{n^2\epsilon^2\mu}{pG^2 \ln(1/\delta)})\right)$,

$$\mathbb{E}[F^r(\tilde{x}_T, D)] - F^r(x_*, D) \leq \tilde{O}\left(\frac{p\log(n)G^2 \log(1/\delta)}{n^2\epsilon^2\mu}\right), \tag{5}$$

where some insignificant logarithm terms are hiding in the $\tilde{O}$-notation. The total gradient complexity is $O\left((n + \frac{L}{\mu})\log\frac{n\epsilon\mu}{p}\right)$.

**Remark 4.2.** We can further use some acceleration methods to reduce the gradient complexity, see [25][3].

## 4.2 Non-strongly convex case

In some cases, $F^r(x, D)$ may not be strongly convex. For such cases, [5] has recently showed that SVRG++ has less gradient complexity than Accelerated Gradient Descent. Following the idea of DP-SVRG, we present the algorithm DP-SVRG++ for the non-strongly convex case. Unlike the previous one, this algorithm can achieve the optimal utility bound.

**Theorem 4.3.** In **DP-SVRG++**(Algorithm 2), for $\epsilon \leq c_1 \frac{2^T m}{n^2}$ with some constant $c_1$ and $\delta > 0$, it is $(\epsilon, \delta)$-differentially private if

$$\sigma^2 = c\frac{G^2 2^T m \ln(\frac{2}{\delta})}{n^2 \epsilon^2} \tag{6}$$

for some constant $c$.

**Theorem 4.4** (Utility guarantee). Suppose that the loss function $f(x, z)$ is convex, G-Lipschitz and L-smooth. In **DP-SVRG++**(Algorithm 2), if $\sigma$ is chosen as in (6), $\eta = \frac{1}{13L}$, and $m = \Theta(L)$ is sufficiently large, then the following holds for $T = O\left(\log(\frac{n\epsilon}{G\sqrt{p}\sqrt{\log(1/\delta)}})\right)$,

$$\mathbb{E}[F^r(\tilde{x}_T, D)] - F^r(x_*, D) \leq O\left(\frac{G\sqrt{p\ln(1/\delta)}}{n\epsilon}\right). \tag{7}$$

The gradient complexity is $O\left(\frac{nL\epsilon}{\sqrt{p}} + n\log(\frac{n\epsilon}{p})\right)$.

---
**Algorithm 2** DP-SVRG++$(F^r, \tilde{x}_0, T, m, \eta, \sigma)$
---
**Input:** $f(x, z)$ is G-Lipschitz, and L-smooth over $x \in \mathcal{C}$. $\tilde{x}_0$ is the initial point, $\eta$ is the step size, and $T, m$ are the iteration numbers.

  $x_0^1 = \tilde{x}_0$
  **for** $s = 1, 2, \cdots, T$ **do**
    $\tilde{v} = \nabla F(\tilde{x}_{s-1})$
    $m_s = 2^s m$
    **for** $t = 1, 2, \cdots, m_s$ **do**
      Pick $i_t^s \in [n]$
      $v_t^s = \nabla f(x_{t-1}^s, z_{i_t^s}) - \nabla f(\tilde{x}_{s-1}, z_{i_t^s}) + \tilde{v} + u_s^t$, where $u_s^t \sim \mathcal{N}(0, \sigma^2 I_p)$
      $x_t^s = \text{prox}_{\eta r}(x_{t-1}^s - \eta v_t^s)$
    **end for**
    $\tilde{x}_s = \frac{1}{m_s} \sum_{k=1}^{m_s} x_k^s$
    $x_0^{s+1} = x_{m_s}^s$
  **end for**
  **return** $\tilde{x}_T$
---

## 5 Differentially Private ERM for Convex Loss Function in High Dimensions

The utility bounds and gradient complexities in Section 4 depend on dimensionality $p$. In high-dimensional (i.e., $p \gg n$) case, such a dependence is not very desirable. To alleviate this issue, we can usually get rid of the dependence on dimensionality by reformulating the problem so that the goal is to find the parameter in some closed centrally symmetric convex set $\mathcal{C} \subseteq \mathbb{R}^p$ (such as $l_1$-norm ball), *i.e.,*

$$\min_{x \in \mathcal{C}} F(x, D) = \frac{1}{n} \sum_{i=1}^{n} f(x, z_i), \tag{8}$$

where the loss function is convex.

[28],[29] showed that the $\sqrt{p}$ term in (5),(7) can be replaced by the Gaussian Width of $\mathcal{C}$, which is no larger than $O(\sqrt{p})$ and can be significantly smaller in practice (for more detail and examples one may refer to [28]). In this section, we propose a faster algorithm to achieve the upper utility bound. We first give some definitions.

---
**Algorithm 3** DP-AccMD$(F, x_0, T, \sigma, w)$
---
**Input:** $f(x, z)$ is G-Lipschitz, and L-smooth over $x \in \mathcal{C}$. $||\mathcal{C}||_2$ is the $\ell_2$ norm diameter of the convex set $\mathcal{C}$. $w$ is a function that is 1-strongly convex w.r.t $|| \cdot ||_\mathcal{C}$. $x_0$ is the initial point, and $T$ is the iteration number.

  Define $\mathcal{B}_w(y, x) = w(y) - \langle \nabla w(x), y - x \rangle - w(x)$
  $y_0, z_0 = x_0$
  **for** $k = 0, \cdots, T-1$ **do**
    $\alpha_{k+1} = \frac{k+2}{4L}$ and $r_k = \frac{1}{2\alpha_{k+1}L}$
    $x_{k+1} = r_k z_k + (1 - r_k) y_k$
    $y_{k+1} = \arg \min_{y \in \mathcal{C}} \{ \frac{L||\mathcal{C}||_2^2}{2} ||y - x_{k+1}||_\mathcal{C}^2 + \langle \nabla F(x_{k+1}), y - x_{k+1} \rangle \}$

    $z_{k+1} = \arg \min_{z \in \mathcal{C}} \{ \mathcal{B}_w(z, z_k) + \alpha_{k+1} \langle \nabla F(x_{k+1}) + b_{k+1}, z - z_k \rangle \}$, where $b_{k+1} \sim \mathcal{N}(0, \sigma^2 I_p)$
  **end for**
  **return** $y_T$
---

**Definition 5.1** (Minkowski Norm). The Minkowski norm (denoted by $|| \cdot ||_\mathcal{C}$) with respect to a centrally symmetric convex set $\mathcal{C} \subseteq \mathbb{R}^p$ is defined as follows. For any vector $v \in \mathbb{R}^p$,

$$|| \cdot ||_\mathcal{C} = \min\{r \in \mathbb{R}^+ : v \in r\mathcal{C}\}.$$

The dual norm of $|| \cdot ||_\mathcal{C}$ is denoted as $|| \cdot ||_{\mathcal{C}^*}$, for any vector $v \in \mathbb{R}^p$, $||v||_{\mathcal{C}^*} = \max_{w \in \mathcal{C}} |\langle w, v \rangle|$.

The following lemma implies that for every smooth convex function $f(x, z)$ which is L-smooth with respect to $\ell_2$ norm, it is $L||\mathcal{C}||_2^2$-smooth with respect to $|| \cdot ||_\mathcal{C}$ norm.

**Lemma 5.1.** For any vector $v$, we have $||v||_2 \leq ||\mathcal{C}||_2 ||v||_\mathcal{C}$, where $||\mathcal{C}||_2$ is the $\ell_2$-diameter and $||\mathcal{C}||_2 = \sup_{x,y \in \mathcal{C}} ||x - y||_2$.

**Definition 5.2** (Gaussian Width). Let $b \sim \mathcal{N}(0, I_p)$ be a Gaussian random vector in $\mathbb{R}^p$. The Gaussian width for a set $\mathcal{C}$ is defined as $G_\mathcal{C} = \mathbb{E}_b[\sup_{w \in \mathcal{C}} \langle b, w \rangle]$.

**Lemma 5.2** ([28]). For $W = (\max_{w \in \mathcal{C}} \langle w, v \rangle)^2$ where $v \sim \mathcal{N}(0, I_p)$, we have $\mathbb{E}_v[W] = O(G_\mathcal{C}^2 + ||\mathcal{C}||_2^2)$.

Our algorithm **DP-AccMD** is based on the Accelerated Mirror Descent method, which was studied in [4],[23].

**Theorem 5.3.** In **DP-AccMD**( Algorithm 3), for $\epsilon, \delta > 0$, it is $(\epsilon, \delta)$-differentially private if

$$\sigma^2 = c \frac{G^2 T \ln(1/\delta)}{n^2 \epsilon^2} \tag{9}$$

for some constant $c$.

**Theorem 5.4** (Utility Guarantee). Suppose the loss function $f(x, z)$ is G-Lipschitz , and L-smooth over $x \in \mathcal{C}$ . In DP-AccMD, let $\sigma$ be as in (9) and $w$ be a function that is 1-strongly convex with respect to $|| \cdot ||_\mathcal{C}$. Then if

$$T^2 = O\left( \frac{L||\mathcal{C}||_2^2 \sqrt{\mathcal{B}_w(x_*, x_0)} n \epsilon}{G \sqrt{\ln(1/\delta)} \sqrt{G_\mathcal{C}^2 + ||\mathcal{C}||_2^2}} \right),$$

we have

$$\mathbb{E}[F(y_T, D)] - F(x_*, D) \leq O\left( \frac{\sqrt{\mathcal{B}_w(x_*, x_0)} \sqrt{G_\mathcal{C}^2 + ||\mathcal{C}||_2^2} G \sqrt{\ln(1/\delta)}}{n\epsilon} \right).$$

The total gradient complexity is $O\left( \frac{n^{1.5} \sqrt{\epsilon L}}{(G_\mathcal{C}^2 + ||\mathcal{C}||_2^2)^{\frac{1}{4}}} \right)$.

# 6 ERM for General Functions

In this section, we consider non-convex functions with similar objective function as before,

$$\min_{x \in \mathbb{R}^p} F(x, D) = \frac{1}{n} \sum_{i=1}^n f(x, z_i). \tag{10}$$

---

**Algorithm 4** DP-GD($x_0, F, \eta, T, \sigma, D$)

---

**Input:**$f(x, z)$ is G-Lipschitz , and L-smooth over $x \in \mathcal{C}$ . $F$ is under the assumptions. $0 < \eta \leq \frac{1}{L}$ is the step size. T is the iteration number.
  **for** $t = 1, 2, \cdots, T$ **do**
    $x_t = x_{t-1} - \eta \left( \nabla F(x_{t-1}, D) + z_{t-1} \right)$, where $z_{t-1} \sim \mathcal{N}(0, \sigma^2 I_p)$
  **end for**
  **return** $x_T$(For section 6.1)
  **return** $x_m$ where $m$ is uniform sampled from $\{0, 1, \cdots, m - 1\}$(For section 6.2)

---

**Theorem 6.1.** In **DP-GD**( Algorithm 4), for $\epsilon, \delta > 0$, it is $(\epsilon, \delta)$-differentially private if

$$\sigma^2 = c \frac{G^2 T \ln(1/\delta)}{n^2 \epsilon^2} \tag{11}$$

for some constant $c$.

## 6.1 Excess empirical risk for functions under Polyak-Lojasiewicz condition

In this section, we consider excess empirical risk in the case where the objective function $F(x, D)$ satisfies Polyak-Lojasiewicz condition. This topic has been studied in [18][27][26][24][22].

**Definition 6.1** ( Polyak-Lojasiewicz condition). For function $F(\cdot)$, denote $\mathcal{X}^* = \arg\min_{x \in \mathbb{R}^p} F(x)$ and $F^* = \min_{x \in \mathbb{R}^p} F(x)$. Then there exists $\mu > 0$ and for every $x$,

$$||\nabla F(x)||^2 \geq 2\mu(F(x) - F^*). \tag{12}$$

(12) guarantees that every critical point (*i.e.,* the point where the gradient vanish) is the global minimum. [18] shows that if $F$ is differentiable and $L$-smooth w.r.t $\ell_2$ norm, then we have the following chain of implications:

Strong Convex $\Rightarrow$ Essential Strong Convexity $\Rightarrow$ Weak Strongly Convexity $\Rightarrow$ Restricted Secant Inequality $\Rightarrow$ Polyak-Lojasiewicz Inequality $\Leftrightarrow$ Error Bound

**Theorem 6.2.** Suppose that $f(x, z)$ is G-Lipschitz, and L-smooth over $x\mathcal{C}$, and $F(x, D)$ satisfies the Polyak-Lojasiewicz condition. In **DP-GD**( Algorithm 4), let $\sigma$ be as in (11) with $\eta = \frac{1}{L}$. Then if $T = \tilde{O}\left(\log(\frac{n^2\epsilon^2}{pG^2\log(1/\delta)})\right)$, the following holds

$$\mathbb{E}[F(x_T, D)] - F(x_*, D) \leq O(\frac{G^2 p \log^2(n) \log(1/\delta)}{n^2\epsilon^2}), \tag{13}$$

where $\tilde{O}$ hides other $\log, L, \mu$ terms.

**DP-GD** achieves near optimal bound since strongly convex functions can be seen as a special case in the class of functions satisfying Polyak-Lojasiewicz condition. The lower bound for strongly convex functions is $\Omega(\min\{1, \frac{p}{n^2\epsilon^2}\})$[6]. Our result has only a logarithmic multiplicative term comparing to that. Thus we achieve near optimal bound in this sense.

## 6.2 Tight upper bound for (non)-convex case

In [34], the authors considered (non)-convex smooth loss functions and measured the utility as $||F(x^{\text{private}}, D)||^2$. They proposed an algorithm with gradient complexity $O(n^2)$. For this algorithm, they showed that $\mathbb{E}[||F(x^{\text{private}}, D)||^2] \leq O(\frac{\log(n)\sqrt{p\log(1/\delta)}}{n\epsilon})$. By using DP-GD( Algorithm 4), we can eliminate the $\log(n)$ term.

**Theorem 6.3.** Suppose that $f(x, z)$ is G-Lipschitz, and L-smooth. In **DP-GD**( Algorithm 4), let $\sigma$ be as in (11) with $\eta = \frac{1}{L}$. Then when $T = O(\frac{\sqrt{L}n\epsilon}{\sqrt{p\log(1/\delta)}G})$, we have

$$\mathbb{E}[||\nabla F(x_m, D)||^2] \leq O(\frac{\sqrt{L}G\sqrt{p\log(1/\delta)}}{n\epsilon}). \tag{14}$$

**Remark 6.1.** Although we can obtain the optimal bound by Theorem 3.1 using DP-SGD, there will be a constraint on $\epsilon$. Also, we still do not know the lower bound of the utility using this measure. We leave it as an open problem.

# 7 Discussions

From the discussion in previous sections, we know that when gradient perturbation is combined with linearly converge first order methods, near optimal bound with less gradient complexity can be achieved. The remaining issue is whether the optimal bound can be obtained in this way. In Section 6.1, we considered functions satisfying the Polyak-Lojasiewicz condition, and achieved near optimal bound on the utility. It will be interesting to know the bound for functions satisfying other conditions (such as general Gradient-dominated functions [24], quasi-convex and locally-Lipschitz in [16]) under the differential privacy model. For general non-smooth convex loss function (such as SVM ), we do not know whether the optimal bound is achievable with less time complexity. Finally, for non-convex loss function, proposing an easier interpretable measure for the utility is another direction for future work.

## Footnotes

*This research was supported in part by NSF through grants IIS-1422591, CCF-1422324, and CCF-1716400.

[2] Bound and complexity ignore multiplicative dependence on $\log(1/\delta)$.

[3] All of the algorithms and theorems in this section are applicable to closed convex set $\mathcal{C}$ rather than $\mathbb{R}^p$.

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
