[Supplementary Material · Appendix.pdf]

# Supplement
# Differential Private Empirical Risk Minimization Revisited: Faster and More General

## 1  Experiments

In this section, we validate our methods using Covertype dataset[1] and logistic regression. This dataset contains 581012 samples with 54 features. We use 200000 samples for training. We compare our **DP-SVRG** algorithm with the **DP-GD** method in [7] for logistic regression with $L_2$-norm regularization.

$$F^r(w, D) = \frac{1}{n} \sum_{i=1}^{n} \log(1 + \exp(1 + y_i w^T x_i)) + \frac{\lambda}{2} ||w||^2,$$

where $\lambda$ is set to be $10^{-2}$.

We also compare our **DP-SVRG++** algorithm with the **DP-GD** method in [7] for logistic regression,

$$F^r(w, D) = \frac{1}{n} \sum_{i=1}^{n} \log(1 + \exp(1 + y_i w^T x_i))$$

We evaluate the optimality gap $\mathbb{E}[F^r(w^{\text{private}}, D)] - F^r(w^*, D)$ and the running time for $\epsilon = \{0.2, 0.5, 1\}$ and $\delta = 0.001$.

From the figure, it is clear that our method outperform the previous results in both cases.

## 2  Details and proofs

### 2.1  Using Advance Composition Theorem to Guarantee $(\epsilon, \delta)$-differential private

As we can see that there are constrains on $\epsilon$ in Theorem 4.1 and Theorem 4.3. The constrains come from Theorem 3.1 (see the proof below). For general $\epsilon$, we can just amplify a factor of $O(\ln(T/\delta))$ on the $\sigma$. However, in this case, we will amplify a factor of $O(\log(Tm/\delta))$ (neglecting other terms) in (5) and (7) in Theorem 4.2 and 4.4; the guarantee of DP is by advanced composition theorem and privacy amplification via sampling [3]. Below we will show this. Consider the i-th query:

$$M_i = \nabla f(x_{t-1}^s, z_{i_t^s}) - \nabla f(\tilde{x}, z_{i_t^s}) + \frac{1}{n} \sum_{i=1}^{n} \nabla f(\tilde{x}, z_i) + \mathcal{N}(0, \sigma^2 I_p),$$

where $i_t^s$ is the uniform sampling. There are $T$-compositions of these queries. By advanced composition theorem, we know that in order to guarantee the $(\epsilon, \delta)$-differential private, we need $(c \frac{\epsilon}{\sqrt{T \log(1/\delta)}}, T/2\delta)$-differential private in each $M_i$ for some constant $c$. Now consider $M_i$ on the

Figure 1: Comparison of DP-SVRG and DP-GD for Logistic regression with different $\epsilon$ and $L_2$-regularization. We set $T = 15, m = 5000$ and use SVRG-BB for step size update in DP-SVRG, $T = 1500$ in DP-GD.

Figure 2: Comparison of DP-SVRG++ and DP-GD for Logistic regression with different $\epsilon$. We set $T = 15, m = 10, \eta = 0.01$ in DP-SVRG++ and $T = 1000, \eta = 0.1$ in DP-GD.

whole dataset (*i.e.,* with no random sample).

$$\tilde{M}_i = \sum_{i=1}^{n} \nabla f(x_{t-1}^s, z_i) - \sum_{i=1}^{n} \nabla f(\tilde{x}, z_i) + \frac{1}{n} \sum_{i=1}^{n} \nabla f(\tilde{x}, z_i) + \mathcal{N}(0, \sigma^2 I_p).$$

From the above, we can see that the $L_2$-sensitive of $\tilde{M}_i$ is $\Delta \leq 2G + \frac{G}{n} \leq 3G$. Thus if $\sigma^2 \geq c_1 \frac{G^2 \log(1/\delta')}{\epsilon'^2}$ for some $c_1$, $\tilde{M}_i$ will be $(\epsilon', \delta')$-differential private. This implies that the query $M_i$ will be $(2\frac{1}{n}\epsilon', \delta')$-differential private, which comes from the following lemma (see Theorem 2.1 and Lemma 2.2 in [3]).

**Lemma 2.1.** If an algorithm $\mathcal{A}$ is $\epsilon'$-differentially private, then for any $n$-element dataset $D$, executing $\mathcal{A}$ on uniformly random $\gamma n$ entries ensures $2\gamma\epsilon'$-differential private.

Let $2\frac{1}{n}\epsilon' = c\frac{\epsilon}{\sqrt{T\log(1/\delta)}}$ and $\delta' = T/2\delta$, that is $\epsilon' = c'\frac{n\epsilon}{\sqrt{T\log(1/\delta)}}$ and

$$\sigma^2 \geq c_2 \frac{GT\log(T/\delta)\log(1/\delta)}{\epsilon^2 n^2}.$$

We can guarantee that $T$ composition of $M_i$ queries is $(\epsilon, \delta)$-differential private.

## 2.2 Proof of Theorem 4.1 and 4.3

*Proof.* W.l.o.g, we assume $G = 1$, *i.e.,* $||\nabla f|| \leq 1$ (otherwise we can rescale $f$).The Proof of Theorem 4.1 and Theorem 4.3 are the same instead of the iteration number (or number of queries). Let the difference data of $D, D'$ be the n-th data. Now, consider the i-th query:

$$M_i = \nabla f(x_{t-1}^s, z_{i_t^s}) - \nabla f(\tilde{x}, z_{i_t^s}) + \frac{1}{n}\sum_{i=1}^{n}\nabla f(\tilde{x}, z_i) + u_t^s, u_t^s \sim \mathcal{N}(0, \sigma^2 I_p),$$

where $i_t^s \in [n]$ is a uniform sample. This query can be thought as the composition of two queries:

$$M_{i,1} = \nabla f(x_{t-1}^s, z_{i_t^s}) - \nabla f(\tilde{x}, z_{i_t^s}) + \mathcal{N}(0, \sigma_1^2 I_p) \tag{1}$$

and

$$M_{i,2} = \nabla F(\tilde{x}, D) + \mathcal{N}(0, \sigma_2^2 I_p) = \frac{1}{n}\sum_{i=1}^{n}\nabla f(\tilde{x}, z_i) + \mathcal{N}(0, \sigma_2^2 I_p) \tag{2}$$

for some $\sigma_1, \sigma_2$. By **Theorem 2.1** in [1] we have $\alpha_{M_i}(\lambda) \leq \alpha_{M_{i,1}}(\lambda) + \alpha_{M_{i,2}}(\lambda)$. Now we bound $\alpha_{M_{i,1}}(\lambda)$ and $\alpha_{M_{i,2}}(\lambda)$.

For $\alpha_{M_{i,1}}$, we can use **Lemma 3** in [1] directly, where $q = \frac{1}{n}$, $f(\cdot) = \nabla f(x_{t-1}^s, \cdot) - \nabla f(\tilde{x}, \cdot)$. For some constant $c_1$ and any integer $\lambda \leq \sigma_1^2 \ln(n/\sigma_1)$, we have

$$\alpha_{M_{i,1}}(\lambda) \leq c_1 \frac{\lambda^2}{n^2\sigma_1^2} + O(\frac{\lambda^3}{n^3\sigma_1^3}). \tag{3}$$

For $\alpha_{M_{i,2}}(\lambda)$, we use the relationship between moment account and Rényi divergence. By Definition 2.1 in [4] we have:

$$\alpha_{M_{i,2}}(\lambda) = \lambda D_{\lambda+1}(P||Q), \tag{4}$$

where $P = \nabla F(\tilde{x}, D) + \mathcal{N}(0, \sigma_2^2 I_p) = \mathcal{N}(\nabla F(\tilde{x}, D), \sigma_2^2)$ and $Q = \nabla F(\tilde{x}, D') + \mathcal{N}(0, \sigma_2^2 I_p) = \mathcal{N}(\nabla F(\tilde{x}, D'), \sigma_2^2)$. By Lemma 2.5 in [4], we have for some $c_2$:

$$\lambda D_{\lambda+1}(P||Q) = \frac{\lambda(\lambda+1)||\nabla F(\tilde{x}, D) - \nabla F(\tilde{x}, D')||^2}{2\sigma^2} \leq \frac{2\lambda(\lambda+1)}{n^2\sigma_2^2} \leq \frac{c_1\lambda^2}{n^2\sigma_2^2}. \tag{5}$$

Combining (3), (4) and (5), we have

$$\alpha_{M_i}(\lambda) \leq c_1\frac{\lambda^2}{n^2\sigma_2^2} + c_2\frac{\lambda^2}{n^2\sigma_1^2} + O(\frac{\lambda^3}{n^3\sigma_1^3}). \tag{6}$$

The rest is similar to the proof of Theorem 3.1.
After $T$ iterations, we have for some $c_1, c_2$,

$$\alpha_M \leq \sum_{i=1}^{T}\alpha_{M_i} \leq c_1\frac{\lambda^2}{n^2\sigma_2^2} + c_2\frac{\lambda^2}{n^2\sigma_1^2}. \tag{7}$$

To be $(\epsilon, \delta)$-differential private, by Theorem 2.2 in [1], it suffices that

$$c_1\frac{T\lambda^2}{n^2\sigma_2^2} + c_2\frac{T\lambda^2}{n^2\sigma_1^2} \leq \frac{\lambda\epsilon}{2}$$

and

$$\exp(\frac{-\lambda\epsilon}{2}) \leq \delta.$$

In addition we need

$$\lambda \leq \sigma_1^2 \ln(n/\sigma_1). \tag{8}$$

It can be verified that when $\epsilon \leq c_3 \frac{T}{n^2}$ for some constant $c_3$, we have

$$\sigma_1 = c_4 \frac{\sqrt{T \log(1/\delta)}}{n\epsilon} \tag{9}$$

and

$$\sigma_2 = c_5 \frac{\sqrt{T \log(1/\delta)}}{n\epsilon}. \tag{10}$$

For some constant $c_4, c_5$, all the conditions can be satisfied. Since the sum of two Gaussian distributions is still a Gaussian distribution, and $M_i = M_{i,1} + M_{i,2}$, we have $\sigma = c \frac{\sqrt{T \log(1/\delta)}}{n\epsilon}$ for some $c$. Thus, T-fold of the queries.

$$M_i = \nabla f(x_{t-1}^s, z_{i_t^s}) - \nabla f(\tilde{x}, z_{i_t^s}) + \frac{1}{n} \sum_{i=1}^{n} \nabla f(\tilde{x}, z_i) + \mathcal{N}(0, \sigma^2 I_p)$$

will guarantee $(\epsilon, \delta)$-differential private when $\epsilon \leq c_3 \frac{T}{n^2}$.
For Theorem 4.1 $T = Tm$ while for Theorem 4.3 $T = 2^{T+1}m$. □

## 2.3 Proof of Theorem 5.3 and Theorem 6.1

*Proof.* The proof is similar to the above.

$$M_i = \nabla F(\tilde{x}, D) + \mathcal{N}(0, \sigma^2 I_p) = \frac{1}{n} \sum_{i=1}^{n} \nabla f(\tilde{x}, z_i) + \mathcal{N}(0, \sigma^2 I_p). \tag{11}$$

By (3) and (4), we have

$$\alpha_{M_i}(\lambda) \leq \frac{2\lambda(\lambda+1)}{n^2 \sigma^2}. \tag{12}$$

Thus, after $T$-iterations, we have for some $c$

$$\alpha_M \leq \sum_{i=1}^{T} \alpha_{M_i} \leq c \frac{T\lambda^2}{n^2 \sigma^2}. \tag{13}$$

Taking $\sigma = c_1 \frac{\sqrt{T \log(1/\delta)}}{n\epsilon}$ for some constant $c_1$, we can guarantee that

$$c \frac{T\lambda^2}{n^2 \sigma^2} \leq \frac{\lambda\epsilon}{2}$$

and

$$\exp(\frac{-\lambda\epsilon}{2}) \leq \delta,$$

which means $(\epsilon, \delta)$-differential privacy due to Theorem 2.2 in [1]. □

## 2.4 Proof of Theorem 4.2

*Proof.* Let $g_t^s = \frac{1}{\eta}(x_{t-1}^s - \text{prox}_{\eta r}(x_{t-1}^s - \eta v_t^s))$. Then we have $x_t^s = x_{k-1}^s - \eta g_t^s$. Thus

$$||x_t^s - x_*||^2 = ||x_{t-1}^s - \eta g_t^s - x_*||^2 = ||x_{t-1}^s - x_*||^2 - 2\eta \langle g_t^s, x_{t-1}^s - x_* \rangle + \eta^2 ||g_t^s||^2. \tag{14}$$

By Lemma 3 in [6], we have the following inequality

$$-\langle g_t^s, x_{t-1}^s - x_* \rangle + \frac{\eta}{2} ||g_t^s||^2 \leq F^r(x_*) - F^r(x_t^s) - \frac{\mu_F}{2} ||x_{t-1}^s - x_*||^2 - \frac{\mu_r}{2} ||x_t^s - x_*||^2$$
$$- \langle v_t^s - \nabla F(x_{t-1}^s), x_t^s - x^* \rangle. \tag{15}$$

Plugging (15) into (14), we have

$$||x_t^s - x_*||^2 \leq ||x_{t-1}^s - x_*||^2 - 2\eta[F^r(x_t^s) - F^r(x_*)] - 2\eta \langle v_t^s - \nabla F(x_{t-1}^s), x_t^s - x^* \rangle. \tag{16}$$

Next we bound $-2\eta\langle v_t^s - \nabla F(x_{t-1}^s), x_t^s - x^*\rangle$. Denote $\hat{x}_t^s = \text{prox}_{\eta r}(x_{t-1}^s - \eta\nabla F(x_{t-1}^s))$.

$$-2\eta\langle v_t^s - \nabla F(x_{t-1}^s), x_t^s - x^*\rangle =$$

$$-2\eta\langle v_t^s - \nabla F(x_{t-1}^s), x_t^s - \hat{x}_t^s\rangle - 2\eta\langle v_t^s - \nabla F(x_{t-1}^s), \hat{x}_t^s - x_*\rangle \tag{17}$$

$$\leq 2\eta||v_t^s - \nabla F(x_{t-1}^s)||||x_t^s - \hat{x}_t^s|| - 2\eta\langle v_t^s - \nabla F(x_{t-1}^s), \hat{x}_t^s - x_*\rangle \tag{18}$$

$$\leq 2\eta||v_t^s - \nabla F(x_{t-1}^s)||||x_{t-1}^s - \eta v_t^s - (x_{t-1}^s - \nabla F(x_{t-1}^s)|| - 2\eta\langle v_t^s - \nabla F(x_{t-1}^s), \hat{x}_t^s - x_*\rangle \tag{19}$$

$$\leq 2\eta^2||v_t^s - \nabla F(x_{t-1}^s)||^2 - 2\eta\langle v_t^s - \nabla F(x_{t-1}^s), \hat{x}_t^s - x_*\rangle \tag{20}$$

The first inequality is due to the following lemma,

**Lemma 2.2.** Let $r$ be a closed convex function on $\mathbb{R}^p$. Then for any $x, y \in \text{dom}(R)$

$$||\text{prox}_r(x) - \text{prox}_r(y)|| \leq ||x - y||.$$

We can easily get $\mathbb{E}_{u_t^s, i_t^s}(v_t^s - \nabla F(x_{t-1}^s) = 0$ since $u_t^s$ is independent with $v_{t-1}^s$. Also by Lemma 1 in [6] and $\mathbb{E}[||a + b||^2] \leq 2\mathbb{E}||a||^2 + 2\mathbb{E}||b||^2$, we have

$$\mathbb{E}_{i_t^s, u_t^s}||v_t^s - \nabla F(x_{t-1}^s)||^2 \leq 8L[F^r(x_{t-1}^s) - F^r(x_*) + F^r(\tilde{x}) - F^r(x_*)] + 2\sigma^2 p. \tag{21}$$

Plugging (20) into (16) and taking the expectation with $i_t^s, u_t^s$, we have

$$\mathbb{E}||x_t^s - x_*||^2 \leq ||x_{t-1}^s - x_*||^2 - 2\eta[\mathbb{E}(F^r(x_t^s) - F^r(x_*)] +$$
$$16\eta^2 L[F^r(x_{t-1}^s) - F^r(x_*) + F^r(\tilde{x}) - F^r(x_*)] + 4\eta^2\sigma^2 p. \tag{22}$$

Summing over $t = 1, 2, \cdots, m$ and taking the expectation, we have

$$\mathbb{E}[||x_m^s - x_*||^2] + 2\eta(1 - 8\eta L)\sum_{t=1}^m[\mathbb{E}(F^r(x_t^s)) - F^r(x_*)] \tag{23}$$

$$\leq ||\tilde{x} - x_*||^2 + 16L\eta^2(m + 1)[F^r(\tilde{x}) - F^r(x_*)] + 4m\eta^2\sigma^2 p. \tag{24}$$

Since $F^r$ is $\mu$ strongly convex, we have $||\tilde{x} - x_*||^2 \leq \frac{2}{\mu}(F^r(\tilde{x}) - F^r(x_*))$. Dividing $2m\eta(1 - 8L\eta)$ from both sides, we get

$$\mathbb{E}[F^r(\tilde{x}^s)] - F^r(x_*) \leq \left(\frac{1}{\eta(1 - 8\eta L)\mu m} + \frac{8L\eta(m + 1)}{m(1 - 8L\eta)}\right)(\mathbb{E}[F^r(\tilde{x}_{s-1})] - F^r(x_*)) + \frac{2\eta}{1 - 8L\eta}\sigma^2 p. \tag{25}$$

Thus we can choose $\eta = \Theta(\frac{1}{L}) < \frac{1}{12L}$ and $m = \Theta(\frac{L}{\mu})$ to make

$$A = \frac{1}{\eta(1 - 8\eta L)\mu m} + \frac{8L\eta(m + 1)}{m(1 - 8L\eta)} < \frac{1}{2}$$

and $\frac{2\eta}{1 - 8L\eta} < \frac{1}{2L}$. By (25) and summing over $s = 1, 2 \cdots, T$ we can get

$$\mathbb{E}[F^r(\tilde{x}^T)] - F^r(x_*) \tag{26}$$

$$\leq A^T[F^r(x_0) - F^r(x_*)] + \frac{\sigma^2 p}{L} \tag{27}$$

$$= A^s[F^r(x_0) - F^r(x_*)] + O(\frac{pG^2 Tm \ln(1/\delta)}{n^2\epsilon^2 L}) \tag{28}$$

$$= A^T[F^r(x_0) - F^r(x_*)] + O(\frac{pG^2 T \ln(1/\delta)}{n^2\epsilon^2 \mu}). \tag{29}$$

Thus if we take T such that $A^T[F^r(x_0) - F^r(x_*)] = O(\frac{pG^2 \ln(1/\delta)}{n^2\epsilon^2\mu})$, i.e.,

$$T = O\left(\log(\frac{n^2\epsilon^2\mu}{pG^2 \ln(1/\delta)})\right).$$

We have

$$\mathbb{E}[F^r(\tilde{x}^T)] - F^r(x_*) \leq O(\frac{pG^2 \ln(n\epsilon\mu/pG) \ln(1/\delta)}{n^2\epsilon^2\mu}).$$

where the big-O notation omitted the other ln term. $\qquad\square$

## 2.5 Proof of Theorem 4.4

*Proof.*

$$\mathbb{E}_{i_t^s,u_t^s}[F^r(x_t^s) - F^r(x_*)] = \mathbb{E}_{i_t^s,u_t^s}[F(x_t^s) - F(x_*) + r(x_t^s) - r(x_*)] \tag{30}$$

$$\leq \mathbb{E}_{i_t^s,u_t^s}[F(x_{t-1}^s) + \langle \nabla F(x_{t-1}^s), x_t^s - x_{t-1}^s \rangle + \frac{L}{2}||x_t^s - x_{t-1}^s||^2 - F(x_*) + r(x_t^s) - r(x_*)] \tag{31}$$

$$\leq \mathbb{E}_{i_t^s,u_t^s}[\langle \nabla F(x_{t-1}^s), x_{t-1}^s - x_* \rangle] + \langle \nabla F(x_{t-1}^s), x_t^s - x_{t-1}^s \rangle$$
$$+ \frac{L}{2}||x_t^s - x_{t-1}^s||^2 + r(x_t^s) - r(x_*)] \tag{32}$$

$$= \mathbb{E}_{i_t^s,u_t^s}[\langle v_t^s, x_{t-1}^s - x_* \rangle] + \langle \nabla F(x_{t-1}^s), x_t^s - x_{t-1}^s \rangle + \frac{L}{2}||x_t^s - x_{t-1}^s||^2 + r(x_t^s) - r(x_*)]. \tag{33}$$

The last equality is due to the fact that $\mathbb{E}_{i_t^s,u_t^s}[v_t^s] = \nabla F(x_{t-1}^s)$. Since we have ([2])

$$\langle v_t^s, x_{t-1}^s - x_* \rangle + r(x_t^s) - r(x_*) \leq \langle v_t^s, x_{t-1}^s - x_t^s \rangle + \frac{||x_{t-1}^s - x_*||^2}{2\eta} - \frac{||x_t^s - x_*||^2)}{2\eta} - \frac{||x_t^s - x_{t-1}^s||^2}{2\eta}. \tag{34}$$

Plugging (34) into (33), we have

$$LHS \leq \mathbb{E}_{i_t^s,u_t^s}[\langle v_t^s - \nabla F(x_{t-1}^s), x_{t-1}^s - x_t^s \rangle - \frac{1-\eta L}{2\eta}||x_t^s - x_{t-1}^s||^2$$
$$+ \frac{||x_{t-1}^s - x_*||^2 - ||x_t^s - x_*||^2}{2\eta}] \tag{35}$$

$$\leq \mathbb{E}_{i_t^s,u_t^s}\frac{\eta}{2(1-\eta L)}||v_t^s - \nabla F(x_{t-1}^s)||^2 + \frac{||x_{t-1}^s - x_*||^2 - \mathbb{E}_{i_t^s,u_t^s}[||x_t^s - x_*||^2]}{2\eta} \tag{36}$$

$$\leq \frac{4\eta L}{1-\eta L}[F^r(x_{t-1}^s) - F^r(x_*) + F^r(\tilde{x}_{s-1}) - F^r(x_*)] + \frac{\eta}{1-\eta L}p\sigma^2$$
$$+ \frac{||x_{t-1}^s - x_*||^2 - \mathbb{E}_{i_t^s,u_t^s}[||x_t^s - x_*||^2]}{2\eta}. \tag{37}$$

Choosing $\eta = \frac{1}{13L}$, summing over $t = 1, \cdots, m_s$, dividing $m_s$, and taking expectation, we have

$$\mathbb{E}[\frac{1}{m_s}\sum_{t=1}^{m_s} F^r(x_t^s) - F^r(x_*)] \leq \frac{1}{3}\mathbb{E}[\frac{1}{m_s}\sum_{t=0}^{m_s-1}[F^r(x_t^s) - F^r(x_*) + F^r(\tilde{x}_{s-1}) - F^r(x_*)] +$$
$$\frac{||x_0^s - x_*||^2 - \mathbb{E}[||x_{m_s}^s - x_*||^2]}{2\eta m_s} + \frac{1}{12L}\sigma^2 p. \tag{38}$$

By the definitions of $x_0^{s+1}$ and $\tilde{x}_s$, we have

$$2\mathbb{E}[F^r(\tilde{x}_s) - F^r(x_*)] \leq \mathbb{E}[\frac{F^r(x_0^s) - F^r(x_*) - (F^r(x_0^{s+1}) - F(x_*))}{m_s} +$$
$$F^r(\tilde{x}_{s-1}) - F^r(x_*) + \frac{||x_0^s - x_*||^2 - ||x_0^{s+1} - x_*||^2}{2\eta/3m_s}] + \frac{1}{4L}\sigma^2 p, \tag{39}$$

which implies that

$$2(\mathbb{E}[F^r(\tilde{x}_s) - F^r(x_*) + \frac{||x_0^{s+1} - x_*||^2}{4\eta/3m_s} + \frac{F^r(x_0^{s+1}) - F^r(x_*)}{2m_s}]) \tag{40}$$

$$\leq \mathbb{E}[F^r(\tilde{x}_{s-1}) - F^r(x_*) + \frac{||x_0^s - x_*||^2}{4\eta/3m_{s-1}} + \frac{F^r(x_0^s) - F^r(x_*)}{2m_{s-1}}] + \frac{1}{4L}\sigma^2 p. \tag{41}$$

Summing over $s = 1, \cdots, T$, we get

$$\mathbb{E}[F^r(\tilde{x}_T) - F^r(x_*)] \tag{42}$$

$$\leq \frac{F^r(\tilde{x}_0) - F^r(x_*)}{2^{T-1}} + \frac{||\tilde{x}_0 - x_*||^2}{2^T 4\eta/3m} + \frac{1}{4L}\sigma^2 p. \tag{43}$$

Thus, if we take $m = \Theta(L)$ to make $A = 2F^r(\tilde{x}_0) - F^r(x_*) + \frac{||\tilde{x}_0 - x_*||^2}{4\eta/3m}$ independent of $T, n, p, \sigma, L$, plug $\sigma$ into (43) we have

$$\mathbb{E}[F^r(\tilde{x}_T)] - F^r(x_*) \leq \frac{A}{2^T} + O(\frac{G^2 p 2^T m \ln 2/\delta}{n^2 \epsilon^2 L}) = \frac{A}{2^T} + O(\frac{G^2 p 2^T \ln(1/\delta)}{n^2 \epsilon^2}). \quad (44)$$

Let $T = O(\log(\frac{n\epsilon}{G\sqrt{p}\sqrt{1/\delta}}))$. We have

$$\mathbb{E}[F^r(\tilde{x}_s)] - F^r(x_*) \leq O(\frac{G\sqrt{p\ln(1/\delta)}}{n\epsilon}).$$

The gradient complexity is $O(2^s m + Tn) = O(\frac{nL\epsilon}{G\sqrt{p}} + n\log(\frac{n\epsilon}{G\sqrt{p}}))$. $\qquad\square$

## 2.6 Proof of lemma 5.1

*Proof.* If $v = 0$, this is true. If not, we will show that $\frac{||v||_2}{||\mathcal{C}||_2} \leq ||v||_{\mathcal{C}}$. This is equivalent to show that $v \notin \frac{||v||_2}{||\mathcal{C}||_2}\mathcal{C}$. Take any $y \in \mathcal{C}$. Since $||\frac{||v||_2}{||\mathcal{C}||_2}y||_2 = \frac{||v||_2}{||\mathcal{C}||_2}||y||_2$, we know that $||y||_2 < ||\mathcal{C}||_2$. Thus $||\frac{||v||_2}{||\mathcal{C}||_2}y||_2 < ||v||_2$. We have $v \notin \frac{||v||_2}{||\mathcal{C}||_2}\mathcal{C}$. $\qquad\square$

## 2.7 Proof of Theorem 5.4

*Proof.* We use $||\cdot||$ and $||\cdot||_*$ instead of $||\cdot||_{\mathcal{C}}$ and $||\cdot||_{\mathcal{C}^*}$. Also, w.l.o.g we assume that $||\mathcal{C}||_2 = 1$ (for the general case, just replace $L$ by $L||\mathcal{C}||_2^2$). Since $b_{k+1}$ is independent of $x_{k+1}$, we have for any $u$

$$\mathbb{E}_{b_{k+1}}[\langle \alpha_{k+1}\nabla F(x_{k+1}), z_k - u\rangle] = \mathbb{E}_{b_{k+1}}[\langle \alpha_{k+1}(\nabla F(x_{k+1}) + b_{k+1}), z_k - u\rangle]$$
$$= \mathbb{E}_{b_{k+1}}[\langle \alpha_{k+1}(\nabla F(x_{k+1}) + b_{k+1}), z_k - z_{k+1}\rangle] + \mathbb{E}_{b_{k+1}}[\langle \alpha_{k+1}(\nabla F(x_{k+1}) + b_{k+1}), z_{k+1} - u\rangle]. \quad (45)$$

Since $z_{k+1} = \arg\min_{z \in \mathcal{C}}\{\mathcal{B}_w(z, z_k) + \alpha_{k+1}\langle \nabla F(x_{k+1}) + b_{k+1}, z - z_k\rangle\}$, which implies that $\langle \nabla \mathcal{B}_w(z_{k+1}, z_k) + \alpha_{k+1}(\nabla F(x_{k+1} + b_{k+1}), u - z_{k+1}\rangle \geq 0$ for every $u \in \mathcal{C}$. So we can get

$$\mathbb{E}_{b_{k+1}}[\langle \alpha_{k+1}(\nabla F(x_{k+1}) + b_{k+1}), z_{k+1} - u\rangle] \quad (46)$$
$$\leq \mathbb{E}_{b_{k+1}}[\langle -\nabla \mathcal{B}_w(z_{k+1}, z_k), z_{k+1} - u\rangle] = \mathbb{E}_{b_{k+1}}[\mathcal{B}_w(u, z_k) - \mathcal{B}_w(u, z_{k+1}) - \mathcal{B}_w(z_{k+1}, z_k)], \quad (47)$$

where the equality is due to the triangle equality of Bregman divergence. Since $w$ is 1-strong convex with respect to $||\cdot||$, we have $-\mathcal{B}_w(z_{k+1}, z_k) \leq -\frac{1}{2}||z_{k+1} - z_k||^2$. Plugging this into (44), we have

$$\mathbb{E}_{b_{k+1}}[\langle \alpha_{k+1}\nabla F(x_{k+1}), z_k - u\rangle] \quad (48)$$
$$\leq \mathbb{E}_{b_{k+1}}[\langle \alpha_{k+1}(\nabla F(x_{k+1}) + b_{k+1}), z_k - z_{k+1}\rangle - \frac{1}{2}||z_{k+1} - z_k||^2] +$$
$$\mathcal{B}_w(u, z_k) - \mathbb{E}_{b_{k+1}}[\mathcal{B}_w(u, z_{k+1})] \quad (49)$$
$$\leq \mathbb{E}_{b_{k+1}}[\langle \alpha_{k+1}\nabla F(x_{k+1}), z_k - z_{k+1}\rangle - \frac{1}{4}||z_{k+1} - z_k||^2] + \alpha_{k+1}^2 \mathbb{E}_{b_{k+1}}[||b_{k+1}||_*^2] \quad (50)$$
$$+ \mathcal{B}_w(u, z_k) - \mathbb{E}_{b_{k+1}}[\mathcal{B}_w(u, z_{k+1})]. \quad (51)$$

The last inequality is due to Cauchy-Shwartz Inequality. Thus we have $\langle \alpha_{k+1}b_{k+1}, z_k - z_{k+1}\rangle \leq \alpha_{k+1}^2||b_{k+1}||_*^2 + \frac{1}{4}||z_k - z_{k+1}||^2$. Now we want to bound $\mathbb{E}_{b_{k+1}}[\langle \alpha_{k+1}\nabla F(x_{k+1}), z_k - z_{k+1}\rangle -$

$\frac{1}{4}||z_{k+1} - z_k||^2$]. Define $v = r_k z_{k+1} + (1 - r_k) y_k \in \mathcal{C}$ so that $x_{k+1} - v = r_k(z_k - z_{k+1})$. We have

$$\langle \alpha_{k+1} \nabla F(x_{k+1}), z_k - z_{k+1} \rangle - \frac{1}{4}||z_{k+1} - z_k||^2 = \langle \frac{\alpha_{k+1}}{r_k} \nabla F(x_{k+1}), x_{k+1} - v \rangle$$

$$- \frac{1}{4r_k^2}||x_{k+1} - v||^2 \tag{52}$$

$$= 2\alpha_{k+1}^2 L(\langle F(x_{k+1}), x_{k+1} - v \rangle - \frac{L}{2}||x_{k+1} - v||^2) \tag{53}$$

$$\leq 2\alpha_{k+1}^2 L(- \min_{y \in \mathcal{C}}\{\frac{L}{2}||y - x_{k+1}||^2 + \langle F(x_{k+1}), y - x_{k+1} \rangle\}) \tag{54}$$

$$= 2\alpha_{k+1}^2 L(-\{\frac{L}{2}||y_{k+1} - x_{k+1}||^2 + \langle F(x_{k+1}), y_{k+1} - x_{k+1} \rangle\}) \tag{55}$$

$$\leq 2\alpha_{k+1}^2 L(F(x_{k+1}) - F(y_{k+1})). \tag{56}$$

The last inequality is due to the fact that $F$ is $L||\mathcal{C}||_2^2$-smooth (note that $||\mathcal{C}||_2 = 1$) in $||\cdot||$ norm and the definition of $y_{k+1}$. Thus, we get the following

$$\mathbb{E}_{b_{k+1}}[\langle \alpha_{k+1} \nabla F(x_{k+1}), z_k - u \rangle] = \mathbb{E}_{b_{k+1}}[\langle \alpha_{k+1}(\nabla F(x_{k+1}) + b_{k+1}), z_k - u \rangle]$$
$$\leq 2\alpha_{k+1}^2 L(F(x_{k+1}) - F(y_{k+1})) + \mathcal{B}_w(u, z_k) - \mathbb{E}_{b_{k+1}}[\mathcal{B}_w(u, z_{k+1})] + \alpha_{k+1}^2 \mathbb{E}_{b_{k+1}}||b_{k+1}||_*^2. \tag{57}$$

By using the Concentration of Gaussian Width, Lemma 3.3 in [5] shows that $\mathbb{E}_{b_{k+1}}||b_{k+1}||_*^2 = \sigma^2 O(G_\mathcal{C}^2 + ||\mathcal{C}||_2^2)$, where $G_\mathcal{C}$ is the Gaussian Width of $\mathcal{C}$. From this, we have

$$E_{b_{k+1}}[\alpha_{k+1}(F(x_{k+1}) - F(u)) \leq \mathbb{E}_{b_{k+1}}[\langle \alpha_{k+1} \nabla F(x_{k+1}), x_{k+1} - u \rangle]$$
$$= \mathbb{E}_{b_{k+1}}[\langle \alpha_{k+1} \nabla F(x_{k+1}), x_{k+1} - z_k \rangle] + \mathbb{E}_{b_{k+1}}[\langle \alpha_{k+1} \nabla F(x_{k+1}), z_k - u \rangle]$$
$$\leq \frac{\alpha_{k+1}(1 - r_k)}{r_k} \langle \nabla F(x_{k+1}), y_k - x_{k+1} \rangle + \mathbb{E}_{b_{k+1}}[\langle \alpha_{k+1} \nabla F(x_{k+1}), z_k - u \rangle]$$
$$\leq \frac{\alpha_{k+1}(1 - r_k)}{r_k}(F(y_k) - F(x_{k+1}) + \mathbb{E}_{b_{k+1}}[\langle \alpha_{k+1} \nabla F(x_{k+1}), z_k - u \rangle]$$
$$\leq (2\alpha_{k+1}^2 L - \alpha_{k+1})(F(y_k) - F(x_{k+1}) + 2\alpha_{k+1}^2 L(F(x_{k+1}) - F(y_{k+1}))$$
$$+ \mathcal{B}_w(u, z_k) - \mathbb{E}_{b_{k+1}}[\mathcal{B}_w(u, z_{k+1})] + \alpha_{k+1}^2 \mathbb{E}_{b_{k+1}}||b_{k+1}||_*^2.$$

Thus we obtain

$$2\alpha_{k+1}^2 LF(y_{k+1}) - (2\alpha_{k+1}^2 L - \alpha_{k+1})F(y_k) + \mathbb{E}(\mathcal{B}_w(u, z_{k+1}) - \mathcal{B}_w(u, z_k)) \tag{58}$$
$$\leq \alpha_{k+1} F(u) + \alpha_{k+1}^2 \sigma^2 O(G_\mathcal{C}^2 + ||\mathcal{C}||_2^2). \tag{59}$$

By the definition of $\alpha_{k+1}$, we have $2\alpha_k^2 L = 2\alpha_{k+1}^2 L - \alpha_{k+1} + \frac{1}{8L}$. Summing over $k = 0 \cdots, T-1$ and setting $u = x_*$, by the definition of $\alpha_k$ we have $\sum_{k=1}^{T} \alpha_k^2 = O(T^3)$. After taking the expectation we get

$$2\alpha_T^2 L\mathbb{E}[F(y_T)] + \frac{1}{8L}\mathbb{E}[\sum_{k=1}^{T-1} F(y_k)] + \mathbb{E}[\mathcal{B}_w(x_*, z_{T-1})] - \mathcal{B}_w(x_*, z_0) \tag{60}$$

$$\leq \sum_{k=1}^{T} \alpha_k F(x_*) + O(T^3 \sigma^2 (G_\mathcal{C}^2 + ||\mathcal{C}||_2^2)/L^2). \tag{61}$$

Plugging $\alpha_k = \frac{k+1}{4L}$ into (59), (60) and dividing both sides by a factor of $2\alpha_T^2 L$, by the fact that $\mathcal{B}_w \geq 0$ we finally get

$$\mathbb{E}[F(y_T)] - F[x_*] \leq \frac{8L\mathcal{B}_w(x_*, x_0)}{(T+1)^2} + O(T\sigma^2 (G_\mathcal{C}^2 + ||\mathcal{C}||_2^2)/L). \tag{62}$$

Since $\sigma^2 = O(\frac{G^2 T \ln(1/\delta)}{n^2 \epsilon^2})$, if choose

$$T^2 = O(\frac{L\sqrt{\mathcal{B}_w(x_*, x_0)}n\epsilon}{G\sqrt{\ln(1/\delta)}\sqrt{G_\mathcal{C}^2 + ||\mathcal{C}||_2^2}}), \tag{63}$$

we have the bound

$$\mathbb{E}[F(y_T)] - F(x_*) \leq O(\frac{\sqrt{\mathcal{B}_w(x_*, x_0)}\sqrt{G_{\mathcal{C}}^2 + ||\mathcal{C}||_2^2}G\sqrt{\ln(1/\delta)}}{n\epsilon}).$$

□

## 2.8 Proof of Theorem 6.2

*Proof.* First of all, we have

$$\mathbb{E}_{z_k}[F(x_{k+1}) - F(x_k)] \leq \mathbb{E}_{z_k}[-\frac{1}{L}\langle \nabla F(x_k), \nabla F(x_k) + z_k \rangle + \frac{1}{2L}||\nabla F(x_k) + z_k||^2] \quad (64)$$

$$= -\frac{1}{2L}||\nabla F(x_k)||^2 + \frac{1}{2L}\mathbb{E}_{z_k}||z_k||^2 \quad (65)$$

$$\leq -\frac{\mu}{L}(F(x_k) - F^*) + \frac{p\sigma^2}{2L}. \quad (66)$$

Re-arranging the terms, we get

$$\mathbb{E}[F(x_{k+1})] - F^* \leq (1 - \frac{\mu}{L})(F(x_k) - F^*) + \frac{p\sigma^2}{2L}.$$

Summing over $k = 0, \cdots, T$ and taking expectation, we obtain

$$\mathbb{E}[F(x_T)] - F^* \leq (1 - \frac{\mu}{L})^T(F(x_0) - F*) + \frac{Tp\sigma^2}{2L}. \quad (67)$$

Thus, when $T = O(\log(\frac{n^2\epsilon^2}{pG^2\log(1/\delta)}))$

$$\mathbb{E}[F(x_T)] - F^* \leq O(\frac{\log^2(n)pG^2\log(1/\delta)}{n^2\epsilon^2}), \quad (68)$$

where the big-$O$ notation neglects other $\log, L, \mu$ terms. □

## 2.9 Proof of Theorem 6.3

*Proof.* The proof is similar to that of Theorem 6.2. Let $F^* = \min_{x \in \mathbb{R}^p} F(x, D)$. We have

$$\mathbb{E}_{z_k}F(x_{k+1}) - F(x_k) \leq \mathbb{E}_{z_k}[-\frac{1}{L}\langle \nabla F(x_k), \nabla F(x_k) + z_k \rangle] + \frac{1}{2L}\mathbb{E}_{z_k}||\nabla F(x_k) + z_k||^2 \quad (69)$$

$$\leq -\frac{1}{2L}||\nabla F(x_k)||^2 + \frac{p\sigma^2}{2L}. \quad (70)$$

From this, we get

$$\frac{1}{2L}||\nabla F(x_k)||^2 \leq F(x_k) - E_{z_k}F(x_{k+1}) + \frac{p\sigma^2}{2L}. \quad (71)$$

Thus, $\mathbb{E}_{m,\{z_i\}}[||\nabla F(x_m)||^2] = \frac{1}{T}\sum_{i=0}^{T-1}\mathbb{E}_{\{z_i\}}[||\nabla F(x_i)||^2]$. By (71), summing over $k = 0, \cdots T - 1$, we obtain

$$\mathbb{E}_{m,\{z_i\}}[||\nabla F(x_m)||^2] \leq \frac{2L(F(x_0) - \mathbb{E}[F(x_T)])}{T}] + p\sigma^2 \quad (72)$$

$$\leq \frac{2L(F(x_0) - F*)}{T} + O(\frac{pG^2\log(1/\delta)T}{n^2\epsilon^2}). \quad (73)$$

Thus, if choose $T = O(\frac{\sqrt{L}n\epsilon}{\sqrt{p\log(1/\delta)}G})$, we have $\mathbb{E}[||\nabla F(x_m)||^2] \leq O(\frac{\sqrt{L}G\sqrt{p\log(1/\delta)}}{n\epsilon})$. □

## Footnotes

[1]https://archive.ics.uci.edu/ml/datasets/covertype