[Reviews · NeurIPS 2017]

Reviewer 1



Summary: The paper revisits the problem of differentially private empirical risk minimization and claims to provide algorithms with tighter gradient complexity (i.e., the number of gradient evaluations to obtain the optimal error). The main algorithm they use is a differentially private variant of the stochastic variance reduced gradient descent (SVRGD) algorithm. Furthermore, they provide excess empirical risk guarantees for non-convex loss functions that satisfy Polyak-Lojasiewicz condition. Positive aspects of the paper: SVRGD has become very popular in the convex optimization literature, and this paper provides the first differentially private variant of it. Furthermore, the analysis for the non-convex case is very interesting. Other comments: i) I believe all the bounds in Table 2 and Table 3 (in terms of gradient complexity) is already known in the literature (up to logarithmic factors). See, the paper "Is Interaction Necessary for Distributed Private Learning?". The main point is that differentially private gradient descent algorithms converge at the same rate as their non-private counter parts up to the optimal error. ii) I am unclear about the Polyak-Lojasiewicz condition. I am sure it is my ignorance of the topic, but the paper does not provide enough intuition into the condition. Given that gradient complexity results are already known, I am worried about the impact of the paper.

Reviewer 2



Summary: A large number of machine learning models are trained on potentially sensitive data, and it is often import to guarantee privacy of the training data. Chaudhuri and Monteleoni formulated the differentially private ERM problem and started a line of work on designing differentially private optimization algorithms for variants of ERM problems. Recent works have gotten nearly optimal tradeoffs between the additional error introduced by the DP algorithm (the privacy risk) and the privacy parameter, for a large class of settings. In this work, these results are improved in the additional axis of computational efficiency. For smooth and strongly convex losses, this work gets privacy risk bounds that are essentially the best known, but do so at a computational cost that is essentially (n+ \kappa) gradient computaitons, instead of n\kappa, where \kappa is the condition number. Similar improvements are presented for other settings of interest, when the loss function is not strongly convex, or when the constraint set has small complexity. A different viewpoint on the results is that the authors show that DP noise addition techniques and modern optimization methods can be made to work well together. Speficially, one can use SVRG with noise addition at each step and the authors show that this noisy SVRG also gets near optimal privacy risk. Similarly for the case of constraint sets with small Gaussian width (such as l_1), where previous work used noisy mirror descent, the authors show that one can use an accelerated noisy mirror descent and get faster runtimes without paying in the privacy cost. I think the problem is very important and interesting. While the tools are somewhat standard, I think this paper advances the state of the art sufficiently that I am compelled to recommend acceptance.

Reviewer 3



This paper gives several algorithm for ERM in convex optimization that satisfy differential privacy. The algorithms improve on known bounds in terms of number of necessary gradient computations and handle some some general settings such as non-convex functions satisfying a certain condition. As far as I can tell from the presentation the results are obtained by plugging the known analyses of gradient perturbation using Gaussian noise into well-known faster algorithms than those previously considered (e.g. SVRG). The stochastic methods naturally deal with randomized estimates of the gradient so accounting for additional randomization due to Gaussian noise is relatively straightforward. These results are useful to have but I think that both technical and conceptual contributions are not quite significant enough for publication in NIPS. (The paper does not contain any discussion of ideas they needed to employ to make the analysis go through so I assume there is not much to discuss). Some minor additional comments: 32. The bound was not "broken" but rather refined using additional structural assumptions. Table 1 caption: L is undefined (I guess it should be the smoothness parameter that you assume to be 1)